

# **Working backwards from streambed thermal anomalies: hydrogeologic controls on preferential brook trout spawning habitat in a coastal stream**

Martin A. Briggs[1*], mbriggs@usgs.gov, (phone) +1.860.487.7402

Judson W. Harvey[2]

Stephen T. Hurley[3]

Donald O. Rosenberry[4]

Timothy McCobb[5]

Dale Werkema[6]

John W. Lane, Jr.[1]

[1]U.S. Geological Survey, Hydrogeophysics Branch, 11 Sherman Place, Unit 5015, Storrs, CT, 06269 USA

[2]U.S. Geological Survey, National Research Program, Reston, VA, 20192 USA

[3]Massachusetts Division of Fisheries and Wildlife, 195 Bournedale Road, Buzzards Bay, MA, 02532 USA

[4]U.S. Geological Survey, National Research Program, M.S. 406, Bldg. 25, DFC, Lakewood, CO, 80225 USA

[5]U.S. Geological Survey, 10 Bearfoot Road, Northborough, MA, 01532 USA

[6]U.S. Environmental Protection Agency, Office of Research and Development, National Exposure Research Laboratory, Exposure Methods & Measurement Division, Environmental Chemistry Branch, Las Vegas, NV, 89119 USA





**Abstract:**
Brook trout (*Salvelinus fontinalis*) spawn in fall, and overwintering egg development
can benefit from stable, relatively warm temperatures in groundwater seepage zones. However,
eggs also are sensitive to dissolved oxygen concentration, which may be reduced in discharging
groundwater. We investigated a 2-km reach of the coastal Quashnet River, Cape Cod,
Massachusetts, USA, to relate preferred fish spawning habitat to geology, geomorphology, and
groundwater discharge. Thermal reconnaissance methods were used to locate zones of rapid
groundwater discharge, which were predominantly found along the center channel of a wider
stream valley section. Pore-water chemistry and temporal vertical groundwater flux were
measured at a subset of these zones during field campaigns over several seasons. Seepage zones
in  open valley sub-reaches generally showed suboxic conditions and higher dissolved solutes
compared to the underlying glacial outwash aquifer. These discharge zones were cross-
referenced with preferred brook trout redds, evaluated during 10 yr of observation, all of which
were associated with discrete alcove features in steep cut banks where stream meander bends
intersect the glacial valley walls. Seepage in these repeat spawning zones was generally stronger
and more variable than open valley sites, with higher dissolved oxygen and reduced solute
concentrations. The combined evidence indicates that regional groundwater discharge along the
broader valley bottom is predominantly suboxic due to the influence of near-stream organic
deposits; trout show no obvious preference for these zones when spawning. However, the
meander bends that cut into sandy deposits near the valley walls generate strong, oxic seepage
zones that are utilized routinely for redd construction and the overwintering of trout eggs. In
similar coastal systems with extensive valley peat deposits, specific use of groundwater
discharge points by brook trout may be limited to morphologies such as cut banks where
groundwater flowpaths can short circuit buried organic material and remain oxygen rich.





## Introduction


The heat tracing of waters can be used to map a distribution of discrete groundwater
discharge zones throughout surface water systems at times of contrast between surface and
groundwater temperature. The measurement of water temperature from the reach to watershed
scale is now possible using thermal infrared (TIR) and fiber-optic distributed temperature
sensing (FO-DTS) methodology (Hare et al., 2015). Remote TIR data collection throughout the
river corridor has been enabled by handheld cameras, piloted aircraft, and the rapidly evolving
capabilities of Unmanned Aircraft Systems. Researchers are capitalizing on the ongoing
refinement of these technologies to identify zones of focused groundwater seepage to streams to
map potential discrete preferential cold-water fish habitat such as summer thermal refugias
(Dugdale et al., 2015). However, surface thermal surveys alone do not indicate groundwater
flowpath dynamics or the suitability of interface aquatic habitat.
For example, dissolved oxygen (DO) concentration must be sufficiently high for cold
groundwater seepage to provide support for fish life-processes at the direct point of discharge to
surface water (Ebersole et al., 2003), which is not apparent from thermal analysis alone. During
summer warm periods in systems with suboxic groundwater, managed cold-water fish species
such as salmonids can face a tradeoff between occupying discrete zones of preferred water
temperatures with near-lethal DO levels, or stream sections that are too warm for long-term
survival (Mathews and Berg, 1997). The use of groundwater upwelling zones as thermal refugia
is further complicated by competition with aggressive invasive species (to the Northeastern
USA) such as brown trout that compete with native trout for resources (Hitt et al., 2017).
Streams at higher elevations may support reach-scale cold water habitat where point-scale
thermal refugia are not needed under current climatic conditions, serving as vital "climate



refugia" against rising air temperatures (Isaak et al., 2015). In systems with reliably cold channel
water in summer, which can also exist at low elevations when heavily influenced by discharging
groundwater, salmonid fish may directly use groundwater seepage zones for spawning rather
than thermal refuge.

Brook trout (*Salvelinus fontinalis*) are a species of char that are native to eastern North

America, from Georgia to Quebec (MacCrimmon and Campbell, 1969). Populations have been
stressed by warming temperatures and reduced water quality, particularly in low-elevation areas
(Hudy et al., 2008). Stream network-scale tracking of fish has indicated brook trout directly
utilize stream confluence mixing zones and groundwater upwelling to survive warm summer
periods (Baird and Krueger, 2003; Petty et al., 2012; Snook et al., 2016). Additionally, brook
trout spawn in the fall, and eggs deposited in redds therefore develop over the winter before
hatching in spring (Cunjak and Power, 1986). Oxygen use by the shallow buried embryos
increases over the period of development (Crisp, 1981), and therefore DO concentration is a
critical parameter of the pore waters in which the eggs are bathed. Several studies have
demonstrated the importance of hyporheic downwelling in increasing shallow oxygen
concentrations specifically at salmonid redds when streambed pore water is generally reduced in
DO (e.g. Buffington and Tonina 2009; Cardenas *et al.* 2016). Fine sediments can reduce the
efficacy of hyporheic DO exchange in spawn zones (Obruca and Hauer, 2016), and are actively
cleared by trout during the spawning process ((Montgomery et al., 1996).

The importance of hyporheic exchange to salmonid spawning may be limited in the

lowland streams that are expected to harbor native cold-water species in the 21st century: those
with strong groundwater influence. Groundwater upwelling reduces the penetration of hyporheic
flow from surface water (Cardenas and Wilson, 2006) and may shut down hyporheic flushing in



redds (Cardenas et al., 2016). Where hyporheic exchange does introduce oxygenated channel
water into the shallow streambed, the downward advection of heat associated with near-freezing
surface water in winter will also cool streambed sediments (Geist et al., 2002), potentially
impairing egg development. Coaster brook trout, a life-history variant of native brook trout
exhibiting potadromous migrations within the Great Lakes, have been shown to specifically
prefer groundwater discharge zones for building redds (Grinsven et al., 2012). The development
of trout in winter has been found to be positively correlated with warmer stream water
temperatures as influenced by groundwater seepage (French et al., 2016), and therefore spatially
discrete groundwater discharge zones with adequate DO may form preferred brook trout
spawning habitat (Curry et al., 1995).

Multiscale physical and biogeochemical factors influence temperature and DO

concentrations along groundwater flowpaths. In river valleys, discharge to surface water of
locally recharged groundwater is expected to emanate from more shallow, lateral flowpaths
controlled by local topography (Modica, 1999; Winter et al., 1998). Deeper regional discharge is
expected to be more vertical through the streambed, as shown using our study site-specific
topography in conceptual Figure 1a. Shallow groundwater flowpaths, particularly those within
approximately 5 m of the land surface, will be more sensitive to annual air temperature patterns
and longer term warming trends due to strong vertical conductive heat exchanges (Kurylyk et al.,
2015). The distance of seeps from upgradient groundwater recharge zones will also affect
seepage temperature dynamics and associated aquatic ecosystems due to future changes in
temperature or precipitation (Burns et al., 2017). Therefore, working backwards from thermal
anomalies into the landscape is critical to understanding the thermal stability of current and
future point-scale preferential brook trout habitat (Briggs et al., 2017a). The complimentary



methodology of geophysical remote sensing, geochemical sampling, and vertical bed
temperature time series can indicate the physical and chemical properties of groundwater
flowpaths that source seepage zones utilized routinely by fish.

Coarse-grained mineral-dominated aquifers with little fine particulate organic matter and

low dissolved organic carbon supply tend to result in generally oxic groundwater conditions
(Back et al., 1993).  The sandy surficial aquifer of Cape Cod, where our investigation took place,
is a good example of a mineral soil-dominated flow system (Frimpter and Gay, 1979). Flow of
groundwater through near-stream organic deposits, however, can result in inverted redox
gradients toward the upwelling interface, such that groundwater discharged to surface water is
reduced in DO (Seitzinger et al., 2006). In sandy glacial terrain with superimposed peatland
deposits, the specific flow patterns of groundwater to surface water in relation to buried peat will
influence groundwater discharge biogeochemistry. Krause *et al.* (2013) found that streambed
groundwater seepage was reduced in DO in zones with peat deposits, likely due to an increase in
both near-stream residence time and localized source of dissolved organic carbon. Whether or
not groundwater flowpaths are dominated by local or regional topography will influence where
and how groundwater discharges to surface water, including possible contact with near-stream
organic deposits (Figure 1a).

Interdisciplinary collaborations between physical and biological scientists are useful to

better understand how cold-water species utilize groundwater seepage-influenced stream habitat,
and the larger landscape-scale controls on seepage zone characteristics. While previous
hydrogeological research in the coastal stream used for this study had focused on locating and
quantifying discrete groundwater discharge (e.g. "cold anomalies", Hare et al., 2015; Rosenberry
et al., 2016), here we endeavor to understand the hydraulic and biogeochemical controls on




seepage zone distribution utilized directly by native brook trout.  In this groundwater-dominated
stream (e.g. likely climate refugia), brook trout do not need to occupy discrete inflows for
summer thermal refugia, but do favor certain upwelling zones for fall spawning. We compare
over a decade of visual and electronic-tag data regarding brook trout redds to a comprehensive
physical and chemical characterization of groundwater seepage zones to:

1. Identify preferred brook trout spawning locations, determine if they are directly

associated with groundwater upwelling, and identify common characteristics (e.g.

temperature, dissolved constituents) between these zones.

2. Develop a hydrogeological understanding of trout-preferred groundwater discharge

zones that can aid in their identification in other less-studied systems.

**Site Description and Previous Ecologic and Hydrogeologic Characterization**

Cape Cod is a peninsula in southeastern coastal Massachusetts, USA, composed

primarily of  highly permeable unconsolidated glacial moraine and outwash deposits. The largest
of the Cape Cod sole-source aquifers occupies a western (landward) section of the peninsula
(LeBlanc et al., 1986), and is incised by several linear valleys that drain groundwater south to the
Atlantic Ocean via baseflow-dominated streams (Figure 2a). Strong groundwater discharge to
one such stream, the Quashnet River, supports a relatively stable flow regime that has averaged
0.49 +/- 0.15 (SD) $m^3 s^{-1}$ from 1986-2015 (Rosenberry et al., 2016). The lower Quashnet River
emerges from a narrow sand and gravel valley to a broader area with well-defined lateral
floodplains. Historical cranberry farming practices, abandoned in the 1950s, have modified the
stream corridor (Barlow and Hess, 1993). Primary modifications included straightening of the
main channel (reducing natural sinuosity), installation of flood-control structures, incision of



shallow groundwater drainage ditches in the lateral peatland floodplain, and widespread
application of sand to the floodplain surface. The current bank-full width of the main channel
averages approximately 4 m.

The Quashnet River has long been recognized as critical habitat for a naturally reproducing

population of native sea-run brook trout (Mullan, 1958) with a genetically distinct population
(Annett et al., 2012).  Efforts to restore trout habitat by the group *Trout Unlimited* and others
have been ongoing for over 40 yr (Barlow and Hess, 1993). These efforts include the removal of
flood-control structures and planting of trees along the main channel, and addition of wood
structures to stabilize banks and provide cover from airborne predators. Further, the
Commonwealth of Massachusetts purchased 31 acres in 1956 and an additional 360 acres along
the lower Quashnet River in 1987 and 1988 to protect the area from development.  The
Massachusetts Division of Fisheries and Wildlife has been monitoring trout populations since
1988 and movement since 2007.

Groundwater influence on stream temperature is pronounced, particularly over the 2-km

reach above the USGS gage, below which stream stage is tidally affected. Temperature
influences in summer include a general downstream cooling with distance toward the USGS
gage (Hare et al., 2015; Rosenberry et al., 2016). Ambient regional groundwater temperature is
approximately 11 °C (Briggs et al., 2014), and strong conductive and advective exchange with
the proximal aquifer maintains surface water temperature well below the lethal threshold for
brook trout (maximum weekly average temperature  >23.3 °C, Wehrly *et al.* 2007). Therefore
niche-scale thermal refugia are not a current concern in this system, as the stream supports
system-scale cold-water habitat that is likely to persist into the future. In winter, seepage zones
can be located as relatively warm anomalies (Hare et al., 2015) increasing and buffering surface



water temperatures from ambient atmospheric influence.

Previous work has measured relatively large net gains in streamflow over the lower Quashnet

River (Barlow and Hess, 1993; Rosenberry et al., 2016), attributed to groundwater discharge
through direct streambed seepage and harvesting of groundwater from the floodplain platform
via relic agricultural drainage ditches. Repeat deployments of fiber-optic temperature sensing
(FO-DTS) cables along the thalweg streambed interface (June 2013, 2014) indicate the greatest
density of focused seepage zones occurs along the broader valley area approximately 1 km
upstream of the USGS gage; this zone coincides with the largest gains in net streamflow (Hare et
al., 2015). Based on the streambed interface temperature data presented by Rosenberry et al.
(2016), Figure 1b shows how temperature-sensitive fiber optic cables have been used to pinpoint
possible groundwater discharge zones based on anomalously cold mean temperature and/or
reduced thermal variance. Focused evaluation of FO-DTS anomalies with physical seepage
meters and vertical temperature profilers confirmed localized, meter-scale seepage zonation
along the streambed (Briggs et al., 2014; Irvine et al., 2016a), where discrete colder zones
indicated through heat tracing showed approximately 5 times the groundwater discharge rate of
adjacent sandy bed locations only meters away (Rosenberry et al., 2016). Active heating of
wrapped FO-DTS cables deployed vertically within an open valley streambed seepage zone
indicated true vertical flow to at least 0.6 m into the bed sediments  (Briggs et al., 2016b), an
expected  characteristic of more regional groundwater discharge (Winter *et al.* 1998; Figure 1a),
rather than that driven by topography local to the river. Hyporheic exchange in the lower
Quashnet River system is superimposed on the general upward hydraulic gradient to the stream,
and therefore reduced to a thin, shallow hyporheic exchange zone (e.g. < 0.1 m depth) along the
thalweg by these competing pressures (Briggs et al., 2014; Rosenberry et al., 2016), as has been



shown for similar systems (e.g. Cardenas and Wilson 2006).
**Methods**

A combination of fish tagging, geophysical surveys, and focused pore-water sampling

was used to investigate the interplay between the locations of preferential brook trout spawning
and the local hydrogeology. For consistency, we adopt the numerical naming convention of
(Rosenberry et al., 2016) for previously identified persistent streambed seepage zones as shown
in Figure 1b. We also refer to the 3 sites of known repeated trout spawning activity as Spawn 1,
2, and 3, from upstream to downstream, respectively (Figure 2).
*Brook trout spatial behavior*

Observations regarding brook trout spawning locations were made as part of an ongoing

PIT (Passive Integrated Transponder) tagging study of the native reproducing population of the
Quashnet River. Large-scale trout movements are continuously monitored in the lower Quashnet
River at 3 stationary fish counting sites (Figure 2a). However the spatial resolution of these
counting sites, separated by hundreds of meters, is not adequate to study how brook trout utilize
specific decimeter- to meter-scale groundwater discharge zones. For this finer scale
characterization, fish tags have also been located through roving surveys using a handheld
portable PIT antenna (Biomark, Inc.) conducted in spring and fall since 2007. In addition to
tagged fish location at the time of these surveys, spawning brook trout were located visually
during fall data collection events and clustering behavior captured within one seepage feature by
underwater video in 2015 using a GoPro Hero camera (San Mateo, CA). Also the dropout of PIT
tags from the fish body is a process that is more likely to happen during spawning behavior in
salmonids (Meyer et al 2011), so dropped tags were located and spatially mapped.



*Fiber-optic distributed temperature sensing*

To augment previously existing streambed interface thermal surveys for groundwater

discharge (e.g. Rosenberry et al., 2016; Figure 1b), ruggedized fiber-optic cables suitable for

stream use were deployed along each bank from approximately 160 m upstream of the middle

fish counter through the Spawn 3 meander bend for approximately 450 m total length (Figure

2a). Two separate cables weighted with stainless steel armoring were installed directly along the

foot of each bank on top of the streambed interface. Single-ended measurements made at the

1.01 m linear spatial sampling scale were integrated over 5-min intervals on each channel by an

Oryx FO-DTS control unit (Sensornet Ltd.). During the same period, data were also collected

along a high-resolution wrapped fiber-optic array for a dataset described in Kurylyk *et al.* (2017)

but not shown here; this experimental setup resulted in measurements for each channel of 4

instrument channels recorded at 20-min intervals. Calibration for dynamic instrument drift was

performed automatically using an approximately 30-m length of cable for each channel

submerged in a continuously mixed ice-bath and monitored with an independent Oryx T-100

thermistor.

*Ground Penetrating Radar*

Ground penetrating radar (GPR) has been successfully applied to several surface

water/groundwater exchange studies to characterize underlying peat and sandy deposits (e.g.

Lowry *et al.* 2009; Comas *et al.* 2011) due to strong expected differences in matrix porosity

(water content), which can exceed 70% in peat (Rezanezhad et al., 2016). An upstream to

downstream GPR profile was collected on July 7, 2016 using a MALA HDR GX160 shielded

antenna (MALA GPR, Sweden) hand-towed down the thalweg from a small inflatable

watercraft. The locations of major seep and spawning sites were marked on the digital GPR



record during data collection. The GPR data were processed using Reflexw software (Sandmeier,
Germany) to convert reflection time to interface depth.
*Temporal groundwater discharge characterization*

Temporal patterns in vertical groundwater discharge flux rate can indicate source

flowpath hydrodynamics, and can be derived from bed temperature time series, as reviewed by
Rau et al., (2013). Custom "1DTempProfilers" designed specifically for the quantification of
groundwater upwelling (Briggs et al., 2014) were used to monitor streambed temperature over
time along a shallow vertical profile. Profilers were deployed in zones of known focused
groundwater discharge and/or preferential trout spawning from June 11 (day 162) to July 13 (day
193) in 2014; August 21 (day 233) to September 13 (day 247) in 2015; and June 5 (day 157) to
July 9 (day 191) in 2016. Individual thermal data loggers (iButton Thermochron DS1922L,
Maxim Integrated) were waterproofed with silicone caulking and inserted horizontally into short
slotted-steel pipes (0.025 m diameter). The shallow thermal profilers were driven vertically into
the streambed so that sensors were positioned at some combination of 0.01, 0.04, 0.07, and 0.11
m depths. Data were collected at temporal intervals of 0.5 hr in 2014, 2015, and 1 hr in 2016.
Rosenberry *et al.* (2016a) found that when a subset of the 2014 streambed temperature data
presented here were analyzed using the diurnal signal amplitude attenuation models employed by
VFLUX2 (Irvine et al., 2015), a near 1:1 relation was found in comparison to physical seepage
meter measurements of groundwater discharge ranging from 0.5 to 3 md$^{-1}$. This strong relation
was likely enabled by using in-situ measurements of thermal diffusivity ($K$e) for modeling as
suggested by Irvine *et al.* (2016) using the diurnal signal phase and amplitude relations presented
by Luce *et al.* (2013). A sequential diurnal signal-based $K$e evaluation to inform amplitude
attenuation-based analytical fluid flux modelling was used here, and this approach is described in



detail by Irvine et al., (2016b).
*Geochemical pore-water characterization*

Subsurface water samples were collected for chemical analysis at ten locations in the

stream along the 2-km study reach using 0.0095 m (nominal) stainless steel drivepoints that had
been inserted to depths of 0.3, 0.6, and 0.9 m.  A 2.4-m length of relatively gas-impermeable
tubing (Masterflex norprene size 15) was attached to the drivepoint and a peristaltic pump was
used to pump groundwater samples until free of obvious turbidity (typically requiring 3 min of
pumping) after which the pumping rate was slowed and, the groundwater samples were collected
by pumping into 60-mL HDPE syringe barrels.  First an unfiltered sample for specific
conductivity was pushed from the syringe into a 30-mL  HDPE Nalgene sample bottle. Second, a
filtered sample for anion analysis was collected after attaching a 0.2-µm pore size (25-mm
diameter) Pall polyethersulfone filter to the syringe. Lastly, the pumping rate was slowed again
and an overflow cup was attached to the norprene sample tubing and held upright until
overflowing, at which point DO was measured by a field colorimetric test using the
manufacturer's evacuated reagent vials, which were snapped inside the overflow cup and then
read on the field photometer (Chemetrics V-2000). DO concentrations were read twice and the
test repeated using an alternative vial kit if results were near the concentration range limit or out
of range. The collected samples were kept cool and out of the light and analyzed for $Cl^-$ upon
return to the laboratory using standard ion chromatographic techniques.

Pore-water samples also were collected from shallow depths ranging between 0.015 and

0.15 m below the streambed surface at the same locations as the drivepoints using minipoint
samplers (e.g. Harvey and Fuller 1998). These small-volume water samples were collected at
slow rates using 0.32-cm stainless steel tubes with slots of 0.01 m forming the screen 0.005 m





behind a clamped tip. The sample tubes were pre-aligned for deployment at selected depths
(0.015, 0.04, 0.08 and 0.15 m) by passing each tube through fittings that gripped the tubes in an
acrylic disc that was lowered until the slotted ends of the sample tubes reached the desired
depths. Water was pumped simultaneously from all depths using a multi-head pump that
withdrew small-volume samples (15 mL) at low flow rates (1.5 mL min$^{-1}$) to minimize
disturbance of natural subsurface fluxes and chemical gradients.  Pumped lines terminated at
press-on luer fittings that were pushed onto 0.2-µm pore size (25-mm diameter) Pall
polyethersulfone filters. Samples for specific conductivity were collected whereas filtered
samples were collected for anions in prelabeled 20-mL LDPE plastic scintillation vials with
Polyseal$^{TM}$ caps. Sample lines were then attached to overflow cups and dissolved oxygen
concentrations were measured as described above.

As mentioned previously, historic cranberry farming practices modified the Quashnet

River valley including the incision of drainage ditches into the floodplain. Some ditches extend
from the valley wall to the main channel, whereas others are shorter or cut at angles. In addition
to characterization of pore water, 34 major drainage ditches (observed flowing water) and a
stream thalweg profile were spot checked for specific conductivity on June, 16 2014 (day 167)
using the SmarTroll probe. At a subset of these ditch locations, filtered grab samples were
collected and analyzed in the laboratory for Cl$^-$ in a similar manner as for the mini and drivepoint
samples described above. In June 2016, the dataset was augmented for 5 ditch confluence
locations upstream of Spawn 1. Also in June 2016, a streambank piezometer was installed on the
hillslope 2.1 m lateral to the Spawn 3 cut bank to a total depth of approximately 3 m and grab
samples were collected after the well was cleared. A basic estimate of Darcy flux to Spawn 3
was made assuming a horizontal gradient, measured at 0.23 compared to stream stage on June, 5



2016 and estimated sand hydraulic conductivity of 10 m/d. Finally, for comparison to Quashnet
River data the characteristic regional groundwater chemical signature of the upgradient
groundwater aquifer was derived from Frimpter and Gay (1979) and Leblanc (1984) for wells
outside of known contaminant influence.
**Results**
Only 3 small alcoves along the 2-km reach were observed to be consistently used for
spawning by brook trout, all of which were associated with meander bend cut banks. Heat
tracing, geophysical, and chemical methods indicate these spawning zones coincide with
localized, oxic groundwater discharge.
*Brook trout spatial behavior*
Out of the dozens of focused seepage zones found along the Quashnet River in this and
previous work (e.g. Figure 1b) brook trout appear to consistently utilize only three zones for
repeat spawning activity. These locations coincide with steep cut banks where the river channel
approaches the sand and gravel valley wall (Figure 2b,c). Specifically, trout were found to
occupy small "scalloped" alcove bank features (Figure 3a) that may be formed by groundwater
sapping and subsequent slumping of sandy bank materials. In winter 2016, fresh slumping and
direct seepage from the newly exposed sand wall was observed at Spawn 3 (Figure 3c); a larger
slump event had filled approximately 1/3 of the scalloped alcove at Spawn 2 by June 2016.
Brook trout were observed clustered along the inner bank area at the Spawn 1 location in fall
2015 (Figure 3d), and this spawning behavior was captured using underwater video
(Supplemental Video S1).
Dropout PIT tags have been located repeatedly in each of the 3 preferential spawn zones.



Seven dropout PIT tags were located in the Spawn 3 zone in March 2017, by far the most
dropped tags found in any one location since the tracking program began in 2007. The only other
obvious persistent scalloped bank features are located at open valley seepage Locations 14/15
(Figure 3b), where Location 14 is near the bank and 15 is in the thalweg. Compared to the trout
spawning zone alcoves (e.g. Figure 3a), this strong open valley alcove was choked with
watercress and thick (tens of centimeters) loose deposits of organic material and spawning trout
have not been observed there.
*Fiber-optic distributed temperature sensing*

The FO-DTS cables deployed at the base of both stream banks through a lower reach

section (Figure 2c) show differing patterns of focused seepage zones indicated by persistent,
cooler anomalies in Figure 4 (Briggs et al., 2017b). The cable along the downstream-right bank
captures a large approximately  8-m-long cooler zone at Spawn 3 (Figure 4b), and this seepage
signature is  spatially reduced but visible along the opposing bank (Figure 4a). Other thermal
anomalies observed along one bank show little or no signature along the other. A short section of
cable (approximately 2 m) was deployed out of the water and over the fish counter apparatus,
and data from this zone show diurnal changes in stream water temperature lag air diurnal
changes by several hours (Figure 4b). Air temperature dropped noticeably over the final 1.5 d of
deployment (day 162), and smaller cool anomalies that appeared on warm days are no longer
captured by the streambed FO-DTS deployment, but the Spawn 3 signature is still visible along
both cables.
*Ground Penetrating Radar*

The GPR data collected along the thalweg adjacent to Spawn 1 and 2 indicate a

contiguous thin layer of material underlies the sandy streambed that may be peat deposited over



deeper sands and gravels (Figure 5a)(Briggs et al., 2017b). The GPR profile through open valley
seepage zone Locations 14/15 and 18 shows the strongest shallow reflectors of anywhere along
the open valley section. These discontinuous interface structures are interpreted as layered sand,
gravel, interspersed with thicker peat deposits (Figure 5b). Otherwise, discontinuous reflections
indicative of sediment type-interfaces of variable depth are observed near downstream open
valley seepage zones where attenuated GPR signals indicate thick lenses of buried peat with high
water content (Figure 5c).
*Groundwater discharge characterization*

Diurnal signal-based $K$e measurements derived from 2 1DTempProfilers inserted in

sandy thalweg sediments for a month in 2014 have the same geometric mean value of 0.11 $m^2d^{-1}$,
and this value is used to model vertical groundwater discharge for all locations and data
collection periods (Briggs, 2017). Upward fluid flux modeling is particularly sensitive to
sediment thermal parameters (Briggs et al., 2014), so reasonable upper and lower bounds of flux
magnitude were estimated as +/- 1 standard deviation of the sub-daily calculations of $K$e
(n=732), or a $K$e range of 0.10-0.13 $m^2d^{-1}$, which is the upper end of the general range observed
for interface sediments (e.g. Rau et al 2012). This uncertainty in thermal parameters could be
expected to generally shift the estimated flux values +/- 0.2 $md^{-1}$ when mean values range 0.5-1.0
$md^{-1}$, and up to  +/- 0.5 $md^{-1}$ for  mean values of 3 $md^{-1}$ (e.g. Spawn 3); however, these shifts do
not impact the general pattern of temporal variability observed primarily at spawn zones.

Sub-daily groundwater discharge fluxes evaluated over similar spring/early summer time

periods in 2014 and 2016 show relatively stable patterns at  open valley seepage zones, generally
<1 $md^{-1}$ (Figure 6a,c). At Spawn 1 and 3 seepage is stronger (2 to 3.5 $md^{-1}$) and more variable
than at open valley zones, with some apparent relation to variations in the stream water stage



evaluated at the USGS gage (Figure 2a). The Darcy-based horizontal seepage estimate through
the Spawn 3 bank, made using the bank piezometer, is 2.3 md$^{-1}$, which is similar to the
temperature-based seepage rates at the Spawn 3 interface (Figure 6c), and indicates lateral
discharge through the cut bank wall from a more localized groundwater flowpath (Figure 1a).
The Spawn 2 zone shows a reduced and more stable discharge rate during summer 2016, and is
likely impacted by a large bank slump into this zone that occurred during the winter of 2016,
partially filling the alcove. Seepage patterns collected at Spawn 1 and 2 in late-summer 2015
show greater temporal stability, even though the stream stage showed substantial variation
(Figure 6b). Discharge rates along the inner wall of the scalloped bank spawn zones were
consistently higher than at bed areas located just a few meters away toward the thalweg (Figure
6a,b).
*Geochemical pore-water characterization*

Based on previous characterization, the regional sand and gravel aquifer generally has

high DO concentrations (9 - 11 mg/L), relatively dilute specific conductance (SpC, 62 µS/cm),
and dilute chloride concentrations (Cl$^-$, 9.3 mg/L) at depths ranging between 12 and 20 m
(Savoie et al., 2012). The groundwater that discharges to the Quashnet, however, is often
strongly variable in all three of these parameters (Harvey et al., 2017), but SpC and Cl$^-$ are used
only to indicate aquifer flowpath properties and not suitable spawn habitat as their range is well
within general brook trout tolerances. In June 2014, drivepoint data were primarily collected in
open valley seepage zones identified with FO-DTS; these locations are generally strongly
suboxic or anoxic at 0.3 and 0.6 m streambed depths (Table 1). The exception is Location 2 in
the tighter upstream valley section, which has a DO concentration of 4.6 mg/L at both depths,
and Spawn 3, where DO is  9.0 and 7.6 mg/L at 0.3 m and 0.6 m depths, respectively. SpC is also





variable, but lowest and similar to the regional signal at Location 2 and Spawn 3.

Drivepoint data collected at the 0.3 m depth in June 2016, primarily around spawn zones,

generally show high DO and relatively low SpC at the interior of Spawn Zones 1 and 3 near the
cut bank (Table 1). Data collected a few meters toward the thalweg from these near-bank spawn
locations are reduced in DO with increased SpC, in an apparent departure from the regional
groundwater signal. The Spawn 2 data were collected at the toe of the recent large sediment
slump that had partially filled the alcove, and DO data are suboxic at 0.3 m (3.9 mg/L) but more
oxygen enriched at 0.9 m depth (7.2 mg/L) indicating the potential for shallow streambed
respiration that removes oxygen from groundwater flow paths (assuming vertical flow). Spawn
Zones 1 and 3 are enriched and reduced in DO at the 0.9 m depth, respectively. In contrast to the
spawn zones, major open valley seepage Locations 14 (near scalloped bank, Figure 3b) and 15
(adjacent thalweg) are nearly anoxic at all depths with SpC similar to the 2014 stream water
profile grab samples (n=8, 101.4 +/- 1.7 µS/cm); little difference was observed between near-
bank and thalweg positions.

The drainage-ditch grab samples generally show $Cl^-$ concentrations that are lower than

the average 2014 thalweg grab samples (n=10, 19 +/- 0.4 mg/L), though the 2 most upstream
ditches are similar to stream water, and 2 open valley ditches are appreciably higher in $Cl^-$
(Figure 7a). Spawn Zones 1, 2, and 3 approximate the lowest $Cl^-$ concentrations observed in
drainage ditches, and Spawn 3 has a similar concentration to the adjacent 2016 streambank
piezometer in both the 2014 and 2016 data. An analogous pattern is shown in the more
widespread SpC data, with many drainage ditches and all spawn zones having concentrations
around 60 µS/cm, but several ditches cluster around the stream water average or higher,
particularly in the open valley area. Concentrations of DO at the drainage ditch confluences were



highly variable, showing no pattern with channel distance, ranging 3.1-8.4 mg/L in June 2014.

The shallow, discrete interval pore-water samples collected with the minipoint system

show that streambed SpC is appreciably lower than stream water, even at the 0.02 m depth, at all
near-bank spawn zones (Figure 8a). Conversely, the shallow thalweg sediments at Spawn 1 and
open valley seepage Location 14/15 approximate the stream water value for SpC. DO is high and
stable along the shallow profiles (to 0.14 m) at the interior of Spawn Zones 1 and 3, suboxic at
the Spawn 1 thalweg and Spawn 2 zones, and essentially anoxic at Location 14 along the entire
profile. Thalweg seepage Location 15 shows moderate oxygen enrichment at 0.02 m (4.6 mg/L),
which may result from hyporheic mixing at the 0.04-0.14 m depths that are nearly anoxic.

Underwater video collected here in the fall of 2015 indicates Quashnet River brook trout

clustered tightly around an approximate 1-m$^2$ bed area in Spawn 1 (Figure 3d, Video S1),
directly at the base of the sandy cut bank. During the June 2016 collection of pore-water data,
drivepoints were installed precisely in this area. Chemical analysis of 0.3 m depth pore water
shows a strong gradient from the near-bank Spawn 1 zone to the outer alcove area, with specific
conductance rising dramatically (70.6 to 143.9 μS/cm) and DO falling (7.28 to 4.41 mg/L)
(Table 1). Spawn 3 shows a similar pattern (60.4 to 82.1 μS/cm SpC; 9.11 to 1.76 mg/L DO),
and Spawn 2, although complicated by the large slump during the previous winter, shows an
increase in SpC from 70.6 to 139.3 μS/cm from the inner to outer alcove. Conversely, pore water
collected at 0.3, 0.6, and 0.9 m depths in the open valley seepage alcove at Location 14/15
(pictured in Figure 3b) are virtually anoxic with elevated SpC compared to inner spawn zones,
and little gradient from bank to thalweg. Fine-scale shallow streambed minipoint data mirror
these deeper samples (Figure 8).





## Discussion


Heat tracing reconnaissance technology, such as FO-DTS and TIR, offer an efficient

means to spatially characterize a subset of focused groundwater discharge points at the reach to
watershed scale (e.g. Figure 1b, Figure 4). Using the groundwater-fed Quashnet River as an
example, Rosenberry et al. (2016) showed that cold streambed interface anomalies in summer
indeed corresponded to discrete zones of particularly high discharge through streambed
sediments. This spatial characterization alone is typically not sufficient to fully understand the
physical and chemical drivers of critical cold-water habitat, but it can greatly focus investigation
of the points of higher-weighted influence on surface water. Compared to more random
streambed field parameter surveys or larger spatial scale evaluations of net groundwater
discharge made with differential gaging, comprehensive spatial mapping of groundwater
discharges is a great advance in the context of understanding point-scale habitat. Here we have
capitalized on previous FO-DTS data collection (Figure 1b) to locate dozens of seepage zones
along a 2-km reach that could be assessed for temporal fluid flux dynamics and chemical
characteristics using subsurface data collection. However, in fast flowing streams even a few
meters wide, cable placement on the streambed will likely impact which specific seepage zones
are captured with FO-DTS, as shown here by applying cables along opposite banks through the
Spawn 3 area (Figure 4). The largest seepage zones may have a spatial footprint that
encompasses the streambed area from bank to bank (e.g., the Spawn 3 cold anomaly), but a
subset of more discrete seepage zones are bound to be missed with a single linear cable
deployment.

In a study of the regional Cape Cod aquifer condition, Frimpter and Gay (1979) state that

groundwater is typically near DO saturation, except downgradient of peat or river bottom




sediments, where consumption of DO allows the mobilization of natural iron and manganese.
Visible observations along the open valley section, in addition to streambed sediment coring
(Briggs et al. 2014), revealed widespread coating of shallow streambed sediment grains with
metal oxides, consistent with the conceptual model of organic material influence on near-surface
groundwater (Figure 1a). Aquifer recharge passing through upgradient groundwater flow-
through kettle lakes (e.g. Stoliker *et al.* 2016) may also serve to decrease the DO content of
regional flowpaths that discharge vertically through the bed of the Quashnet River, although we
hypothesize that localized peat deposits may be the primary control on both seepage zone
distribution and chemistry.

Out of the dozens of focused seepage zones located along the lower Quashnet with heat

tracing, most were suboxic to anoxic (Tables 1,2). Brook trout seem to consistently prefer 3 areas
for fall spawning, all along meander bend cut banks into the sand and gravel valley wall. Zones
of  locally enhanced seepage, likely controlled by subtle differences in sediment hydraulic
conductivity, can lead to groundwater sapping of fines, reduction in bank stability, and
consequent slumping of bank material into the river; this process was observed in real-time at the
Spawn 3 meander in February 2016 (Figure 3c). Slumping effectively forms *seepage-driven*
alcoves outside of the main flow along banks where bed shear stress is reduced and more suitable
for redd placement, along with a more favorable course sand and gravel substrate (Bowerman et
al., 2014; Hausle and Coble, 1976; Raleigh, 1982).

In other systems, trout have been observed to occupy microhabitat around and within

groundwater discharge zones, even segregating by fish size and desirable temperature range
(e.g., Figure 2.4.1.2 in Torgersen et al. 2012). Here real-time observation and visual imagery
show trout clustering tightly against the bank in Spawn 3 (Figure 3d, Video S1), where pore



water was found to be more oxygen rich and lower in SpC. Month-long time series of bed
temperature-derived fluid flux show that the vertical groundwater discharge rate is reduced
considerably from inner to outer alcove zones, indicating a strong reduction in hydraulic gradient
and/or decrease in effective streambed hydraulic conductivity. The evidence of higher near-bank
vertical groundwater flux rates and DO, combined with lower SpC, indicates limited interaction
between shallow groundwater flowpaths and peat against the meander bend cut banks, resulting
in groundwater discharge zones most preferred by brook trout for spawning. The remote sensing
of streambed material with GPR indicates a relatively thin layer of streambed peat in the Spawn
1 and 2 thalweg area compared to open valley seepage zones (Figures 1 and 5). Therefore, it
appears that even short travel distances through organic deposits toward the center channel at
Spawn 1 and 2 may be sufficient to increase total dissolved solids and deplete DO, as observed
in other systems (e.g. Levy et al., 2016), and render upwelling zones undesirable for redd
construction. This characterization is consistent with previous GPR data collected several
kilometers upstream in a broad valley area in a study to assess possible naturalized channel form
restoration (personal communication Maggie Payne, John Cody, Melissa Kenefick, Natural
Resources Conservation Service, November 30, 2015). Their assessment found peat deposits >5
m thick in the central area of the valley, pinching out against the sands and gravels of the valley
walls. Cored peat sections were indicative of a buried cedar swamp, which is typical of similar
glacial depressions in the area (Hare et al., 2017).
Only where seepage was observed to emanate directly from the valley wall sands and
gravels, such as the newly exposed slump in Figure 3C, may groundwater discharge reliably
support overwinter trout egg development. These features are apparently similar to the numerous
cold-water alcove patches observed by Ebersole et al. (2003). In that study of preferential



salmonid habitat, alcoves were often located where streams converged on valley walls and were
the most abundant type of discrete cold-water habitat type identified. Conversely, valley wall
alcoves were the least-common type of seep morphology observed along the Quashnet River. It
is likely that the artificial reduction in channel sinuosity along the Quashnet River through
farming practices has reduced possible higher-quality spawning locations by focusing river flow
away from the valley walls and oxic groundwater discharge. Other bank alcove features with
strong groundwater discharge found along the open valley section (Figure 3b) were highly
influenced by organic material deposition and did not apparently support spawning habitat. Our
research indicates that in lowland systems with organic-rich floodplain sediments (e.g. Figure
1a), valley wall alcoves create favored brook trout spawning habitat via mineral soil-dominated
groundwater discharge flowpaths. It seems reasonable to infer that these features would also
create preferential thermal refugia in streams at times when main channel water exceeds fish
thermal tolerances.

The pore-water chemical data alone do not definitively show that seepage at the cut bank

spawn sites is derived from more localized groundwater recharge, as opposed to regional
groundwater that is unadulterated by buried peat lenses. However, the hydrodynamic data
derived from long-term vertical temperature profiling in seepage zones does offer additional
insight. In general, groundwater discharge rates are more variable at cut bank spawn zones than
in the open valley streambed zones (Figure 6a,c), and this variability may be tied to shorter-term
changes in local river stage and/or water table depth, impacting the hydraulic gradient. The
temporally consistent patterns of open valley discharge may be controlled by the regional
gradient where the flowpath-length term dominates, rendering the Darcy gradient relatively
insensitive to discharge zone changes in river stage or bank proximity. Previous active streambed





heating experiments have indicated open valley seepage to be vertical in nature to >0.6 m depth
also indicating regional discharge (Briggs et al., 2016a), compared to lateral local discharge
through the steep cut bank, indicated by the bank piezometer-stream stage lateral gradient at
Spawn 3.

Groundwater drainage-ditch data collected along the river corridor indicate low SpC/Cl$^-$

conditions exist for the majority of ditches throughout the lower Quashnet River riparian areas
(Figure 7). The hillslope piezometer in sand and gravel at the down valley wall has a similar
chemical signature along with high DO. This similarity is further indication that low-SpC
groundwater discharges even to the lower portion of the river corridor, but is predominantly
modified chemically by travel through near-stream organics. The relic drainage ditches allow
discharging groundwater to effectively short circuit the valley floor peat deposits and remain
high in DO, similar to the natural valley wall cut bank alcoves. Future restoration strategies in
similar systems may consider capitalizing on this short circuit behavior, possibly by auguring
through buried streambed peat or movement of the stream channel toward the valley wall to
create more desirable brook trout aquatic habitat.
**Conclusions**

The three preferential spawn zones that have been identified over 10 yr+ of observation

in the 2-km study reach have strongly discharging groundwater with high DO concentration. The
zones are located exclusively in side alcoves of the channel created by bank slumps along
meanders where the river cuts into steep hillslopes along the glacial sands and gravel valley wall
(Figure 1a). In the alcoves at the base of the cut banks, hillslope groundwater with high DO
concentration is discharged through the streambed without appreciable loss of oxygen. Just a few





meters away toward the main channel, however, groundwater consistently discharges at lower
rates, is reduced in DO, and increased in SpC. The lowest oxygen concentrations in groundwater
are associated with water emerging from the streambed adjacent to wide riparian areas that flank
the Quashnet in the open valley section of the study reach. In the open valley zone, where the
stream is not near the valley walls, proximity to the stream bank does not seem to control
seepage chemistry, and GPR data indicated thick zones of discontinuous peat. In this and other
groundwater-dominated streams that are expected to serve as climate refugia for future native
trout populations, hyporheic exchange will be limited by strong upward hydraulic gradients, and
preferential spawning habitat may be primarily supported by discrete zones of oxic groundwater
upwelling at the meter to sub-meter scale as has been indicated by previous work (e.g. Curry et
al., 1995).

In systems where all groundwater discharge is anoxic, preferential salmonid spawning

zonation may be controlled by points of downwelling hyporheic water where shallow sediments
remain high in DO (Buffington and Tonina, 2009; Cardenas et al., 2016). However, these
hyporheic areas will deliver cold surface water to shallow sediments during winter, which may
impair overwintering brook trout eggs (French et al., 2016). Here, and in many other coastal
systems, groundwater temperature is expected to range approximately 10-12 $^{o}$C, which is an
ideal range for egg development (Raleigh, 1982). Points of oxic groundwater upwelling devoid
of near-stream buried organics, combined with a recirculating side alcove and favorable sand and
gravel sediments, may provide ideal and unique groundwater seepage-enabled preferential
spawning habitat for native trout.

Stream surface or streambed interface heat tracing of groundwater discharge offers an

efficient means to locate discrete seepage zones, but offers only limited insight into the



groundwater hydraulics and biogeochemistry that impact localized trout habitat. A combined
heat tracing and valley-scale geomorphic assessment may be needed to locate probable
preferential seepage zones in other glacial systems, and guide stream ecological restoration
design. As digital elevation models become more refined and combined with infrared data
derived from Unmanned Aircraft Systems, remote identification of relatively small features such
as the seepage alcoves described here should be possible.
**Acknowledgements**

Comments from anonymous reviewers and U.S. Geological Survey (USGS) reviews by

Nathaniel Hitt and Paul Barlow are gratefully acknowledged. The U.S. Environmental Protection
Agency (USEPA) through its Office of Research and Development partially funded and
collaborated in the research described here under agreement number DW-14-92381701 to the
USGS. The USGS authors were supported by the following USGS entities: Office of
Groundwater, Water Availability and Use Science Program, National Water Quality Program,
and the Toxics Substances Hydrology Program. Field and laboratory assistance from Allison
Swartz, Jay Choi, Jenny Lewis, Yao Du, Danielle Hare, Courtney Scruggs, Rayna Mitzman,
David Rey, Geoff Delin, Eric White, MassWildlife Southeast District Staff, Jennifer Salas, and
volunteers from Trout Unlimited is greatly appreciated. The manuscript has been subjected to
Agency review and approved for publication. The views expressed in this article are those of the
authors and do not necessarily represent the views or policies of the USEPA. Any use of trade,
firm, or product names is for descriptive purposes only and does not imply endorsement by the
U.S. Government.





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





**Tables**
Table 1. 2014 and 2016 drivepoint pore-water chemistry data collected in major streambed
groundwater seepage zones located with fiber-optic heat tracing, and in zones of observed repeat
trout spawning directly along the bank ("in") and farther toward the stream thalweg ("out").

| June 2014 Location | 0.3 m depth | | 0.6 m depth | |
|---|---|---|---|---|
| | DO mg/L | SpC μS/cm | DO mg/L | SpC μS/cm |
| Location 2 | 4.6 | 53.8 | 4.6 | 61.3 |
| Location 4 | 1.4 | 97.7 | 3.4 | 65.1 |
| Location 15 | 0.1 | 78.8 | 0.0 | 82.5 |
| Location 18 | 0.2 | 100.0 | 0.2 | 89.8 |
| Location 21 | 0.0 | 77.7 | 0.0 | 79.0 |
| Location 24 | 0.1 | 69.1 | 0.0 | 64.3 |
| Location 27 | 1.4 | 75.0 | 0.5 | 79.4 |
| **Spawn 3 in** | **9.0** | **56.4** | **7.6** | **60.9** |
| June 2016 Location | 0.3 m depth | | 0.9 m depth | |
| **Spawn 1 in** | **7.28** | **70.6** | **9.76** | **55.9** |
| Spawn 1 out | 4.41 | 143.9 | 5.68 | 143.2 |
| **Spawn 2 in** | **3.89** | **70.8** | **7.17** | **57.6** |
| Spawn 2 out | 5.25 | 139.3 | n/a | n/a |
| **Spawn 3 in** | **9.11** | **60.4** | **4.91** | **71.9** |
| Spawn 3 out | 1.76 | 82.1 | 2.68 | 79.9 |
| Location 15 in | 0.16 | 105.5 | 0.39 | 104.0 |
| Location 15 out | 0.31 | 99.1 | 0.18 | 96.4 |




**Figure List**

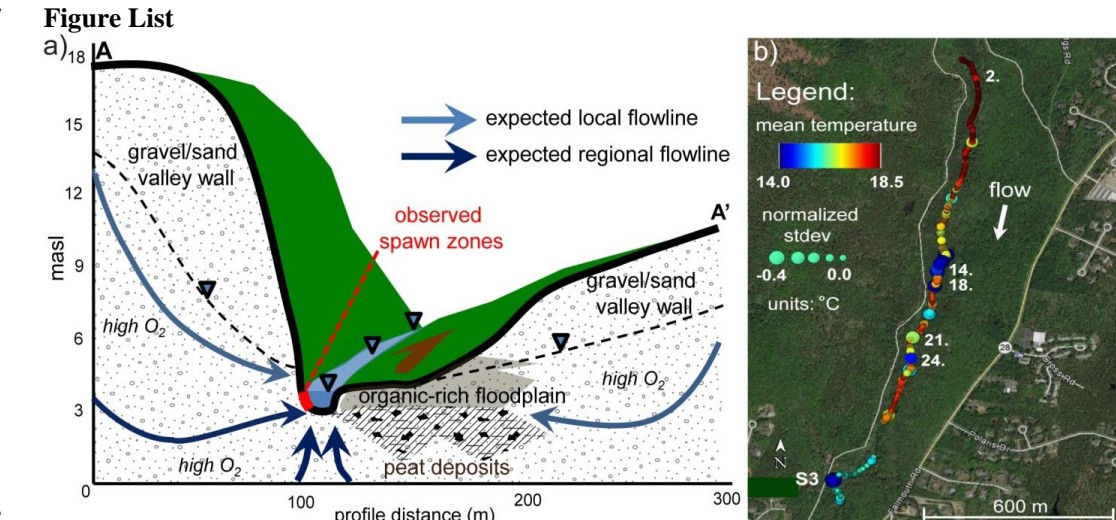

Figure 1. Based on previously published USGS work (e.g. Modica, 1999; Winter et al., 1998),
the conceptual model in panel a) displays how groundwater discharge to lowland streams is
expected to include locally sourced lateral groundwater discharge through valley wall features
and more regionally sourced groundwater discharge vertically through the streambed. The
topographic profile shown here (A-A') is derived directly from lidar data in the vicinity of
observed preferential brook trout spawning habitat shown in Figure 2. In contrast to the sand and
gravel valley walls, multiple methodologies used for this study indicate wider valley zone
sediments to be rich in organic material, including buried peat deposits, consistent with known
regional geology. Modified in format from Figure 4 in Rosenberry et al. (2016), panel b) shows
summary fiber-optic-distributed temperature data collected at 1-m scale along the streambed
interface. Larger dots indicate reduced thermal variance as buffered by groundwater discharge,
while points of relatively cold mean temperatures also suggest strong groundwater influence.
Discharge zones explored in detail are noted numerically following the nomenclature of
Rosenberry et al. (2016).



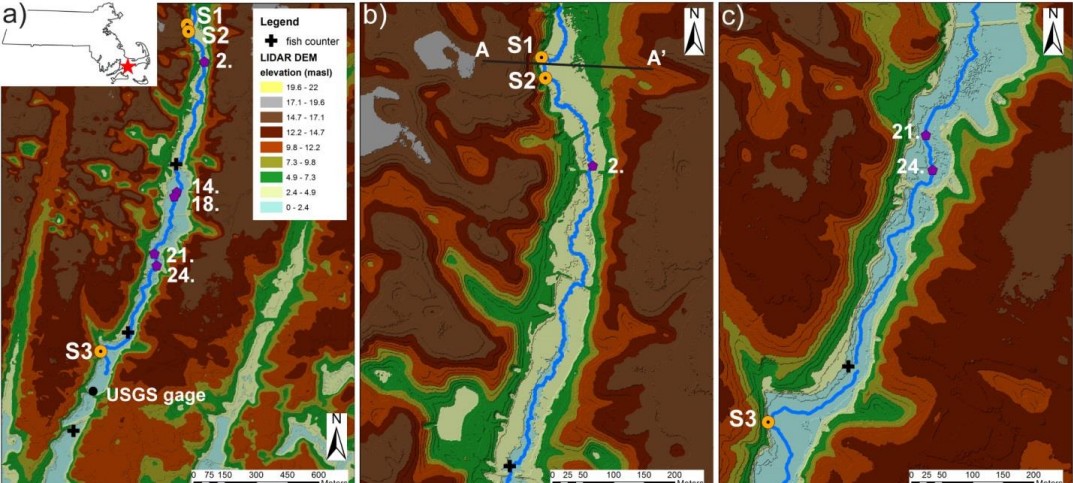

Figure 2. Lidar elevation data show the linear valley terrain of: a) The lower Quashnet River
reach with Spawn (S1, S2, S3) locations (orange circles) and open valley seepage zones (purple
circles)  identified. Panel b) shows the tighter upper valley zone where Spawn 1 and 2 are located
at the base of a steep cut bank and the topographic transect of Figure 1 is noted.  Panel c)
displays the lower open valley reach where Spawn 3 is located along a cut bank.



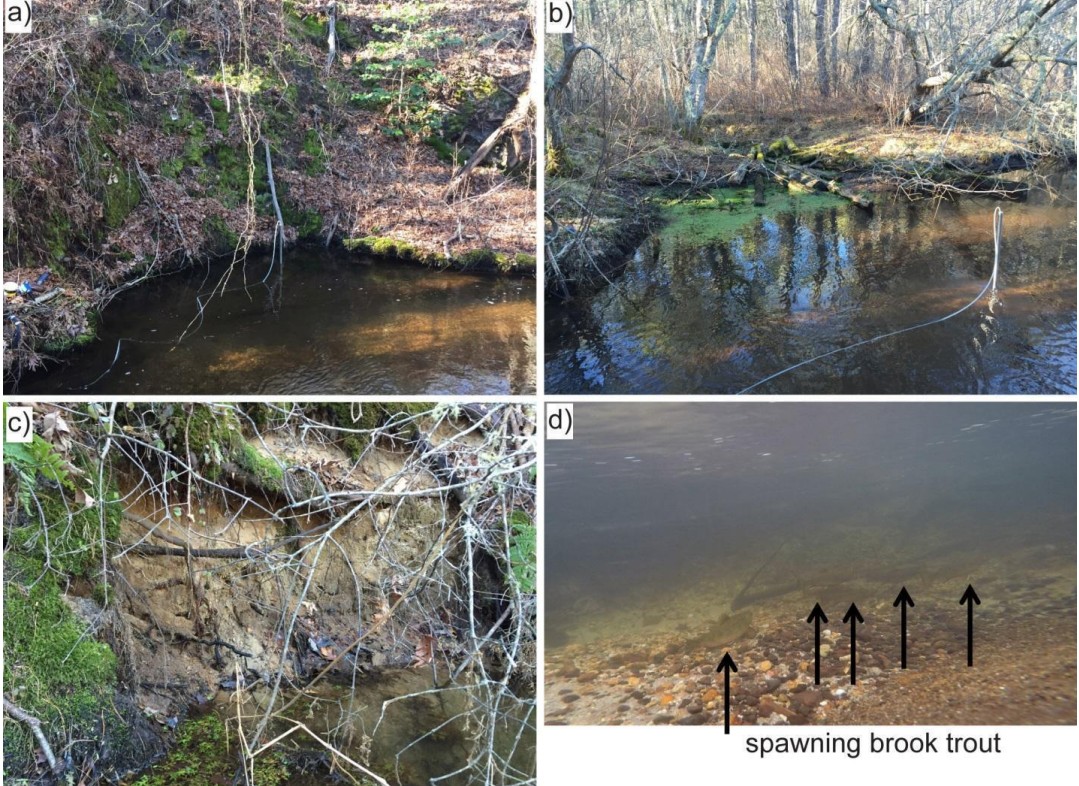

Figure 3. Images collected in February 2016 a) the cut bank alcove at Spawn 1, b) open valley
seepage Location 14/15, and c) fresh cut bank slumping and visible seepage Spawn 3. Panel d) is
an image from the underwater video collected in fall 2015 of spawning trout in the alcove
pictured in panel a), showing several fish clustered around the sandy zone directly at the base of
the cut bank.



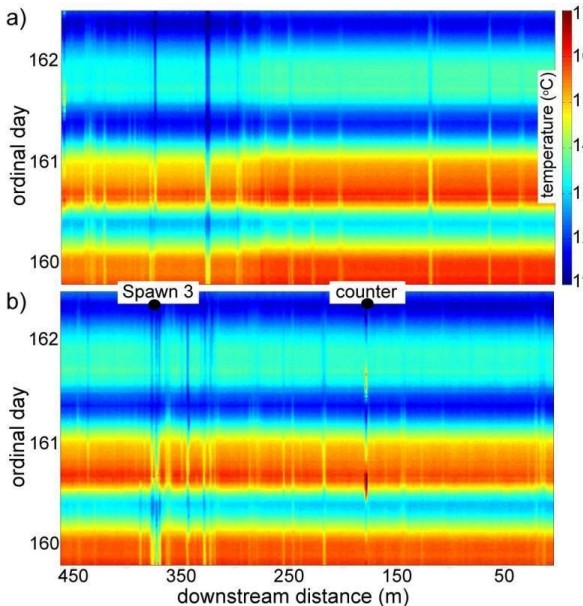

Figure 4. Fiber-optic-distributed temperature data collected starting at an arbitrary stream point
along a) the downstream left bank and b) the downstream right bank through the Spawn 3
meander bend area (see Figure 2c for location). The vertical bands of cooler (blue) colors
indicate discrete groundwater discharge; some larger zones display a thermal signature on both
banks while smaller discharges may be bank-specific.




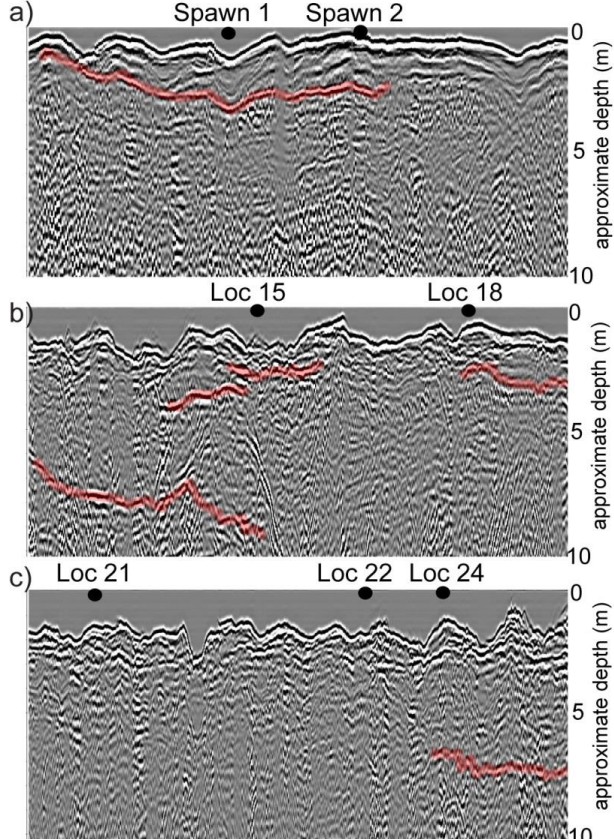

Figure 5. Quashnet River thalweg ground penetrating radar profiles were collected in the vicinity
of: a) Spawn 1 and 2; b) open valley seepage Locations 15 and 18; and c) open valley seepage
Locations 21, 22, and 24. Stronger apparent reflectors are highlighted in red, and likely indicate
sediment layer boundaries (e.g. sand/gravel and peat).



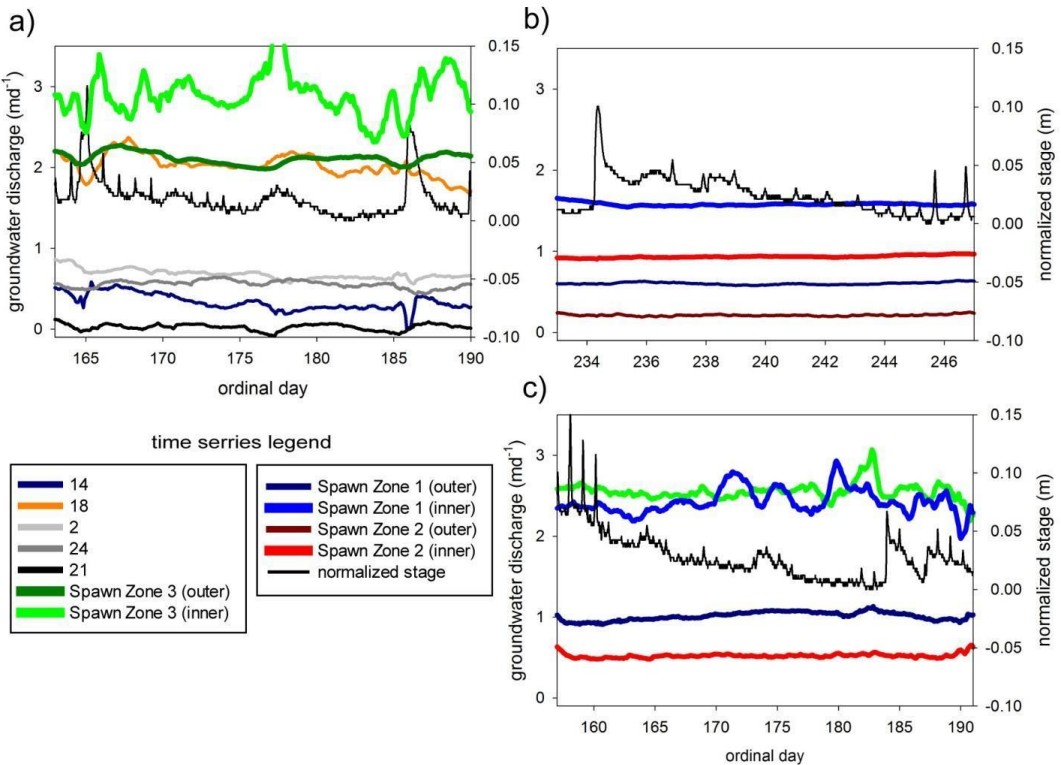

Figure 6. The Quashnet River USGS gaging station stage is compared to sub-daily time series of
vertical groundwater flux rate at various strong seepage zones in a) June 2014, b) August 2015,
and c) June 2016.

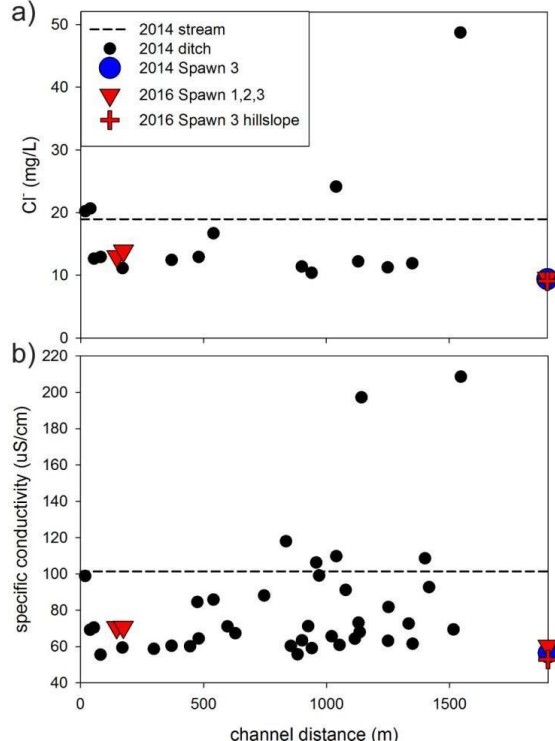

Figure 7. Drainage ditch chemistry throughout the lower Quashnet showing a) Cl⁻, and b)
specific conductance, collected in June 2014 just above the confluence with the main channel.
Data are plotted as distance from the upper flood control structure in the narrow valley reach and
compared to groundwater seepage data collected in preferential spawning locations and a
hillslope piezometer.





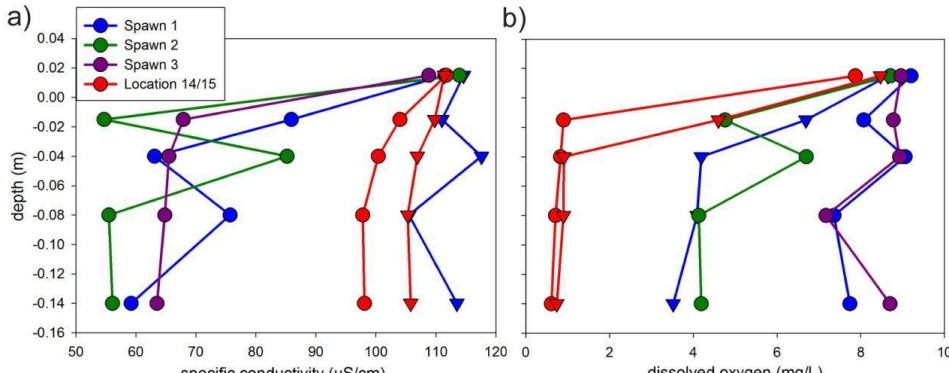

Figure 8. Minipoint pore-water chemistry data showing high spatial resolution profiles of a)
specific conductance, and b) dissolved oxygen, collected in June 2016 at the major seepage
alcoves. Triangle symbols indicate data collected farther toward the thalweg from the respective
alcove bank, and all profiles include a local stream water sample taken just above the streambed
interface.
**Supplemental**
Supplemental Video S1. Underwater video of brook trout spawning in the fall of 2015 (still
image Figure 3d).