# Peer review of "Hydrogeochemical Controls on Brook Trout Spawning Habitat in a Coastal Stream Martin A. Briggs1\*, mbriggs@usgs.gov, (phone) +1.860.487.7402 Judson W. Harvev2 Stephen T. Hurley3 Donald O. Rosenberry4 Timothy McCobb5 Dale Werkema6 John"

_Hydrology and Earth System Sciences, 2017_

## Referee Comment (RC1) · Anonymous Referee #1 · 17 Feb 2018

General comments

The manuscript by Briggs et al. is an impressive multidisciplinary tour de force on groundwater and surface water processes and patterns and trout spawning distribution. The value of this work is in the many different field-based approaches that were used to characterize the thermal, geologic, hydrologic, and ecological environment of the 2-km reach of interest. This strength, however, is also a weakness in that there is so much described in the manuscript that it is difficult to determine the degree to which the authors truly address the objectives set forth at the end of the introduc-

tion. Certainly, the authors "develop a hydrogeological understanding" in relation to the spawning behavior of coldwater fish, but strictly speaking, the actual results presented do not evaluate associations between trout spawning locations and identify common characteristics of these areas. In this sense, to address objective 1 (line 156), the authors would need to statistically analyze the data (i.e., with consideration of sample size and probability) in order to test their hypotheses rather than simply presenting descriptive graphs. As mentioned above, however, the manuscript is interesting in its totality, so perhaps the best approach is to modify the objectives (mainly objective 1) so that they better match what is actually presented in the manuscript. Given the broad focus of this journal and the kinds of articles that it publishes, it seems that the paper is valuable as a "perspective" and "approach" rather than a traditional scientific paper that tests specific hypotheses. This kind of focus is apparent in the manuscript's title, which is good in the images that it evokes, but it is rather vague and would benefit from more explicit wording as opposed to "working backwards", which is rather difficult to understand unless one reads the whole manuscript. In conclusion, some general reframing of the manuscript would be helpful to highlight what it is actually about. This could be accomplished by revising the title and objectives and potentially merging the "conclusions" with the discussion because strictly speaking, this is not a paper that has conclusions in a traditional sense that are derived from statistical analyses.

Specific comments

1. Line 47: Using the word "preferred" is problematic because this paper does not actually statistically analyze preference which would require an explicit comparison to what is available. I recommend removing the word "preferred" from the manuscript. 2. The manuscript refers to 10 years of observation that have gone into counting and mapping redds and spawning behavior, but these data (and sample sizes, etc.) are never presented. Perhaps the authors refer to a manuscript that published these data. In any case, it would be better to temper this kind of wording so that the readers won't be expecting some kind of explicit analysis of these data. 3. Line 58: In many places,

words and phrases are used that are colloquial and informal (e.g., short circuit, dropout, choked, etc.). Although I understand what the authors mean, many readers would not understand these phrases, so it would be best to go through the whole manuscript and eliminate all of this informal language. 4. These are minor issues, but we refer to "coldwater and warmwater fishes" not "cold-water fish. It's just convention that this wording is recommended by the fisheries community (American Fisheries Society). In this sense, it is different from "cold-water seep", for example, which is a hydrologic rather than ecological feature. Also, please be careful about referring to refugia, which is the plural form of refuge or refugium. Thus, we don't write "refugias". 5. Line 182: Shouldn't this be in hectares? 6. Line 193: The word "niche" does not have a scale per se, so it isn't appropriate to use it this way. Be more explicit about what scale you are writing about. 7. Line 236: No information is provided on the number of fish that were tagged. As outlined in my general comments, it would be better to keep the discussion of fish general. 8. Line 343: Can you cite the previous work on the distribution of brook trout spawning? Otherwise, this can't really be presented as a result because we have no data to evaluate in relation to this statement. 9. Figure 1b: This is really too small to examine and appreciate. It's really impressive, but honestly the dots are too busy and crowded. 10. Figure 7: The red and blue symbols are really hard to see on the graph because they overlay one another. Can you jitter them a little so that it is easier to see them?

---

## Referee Comment (RC2) · Anonymous Referee #2 · 19 Feb 2018

General comments:

The paper presents an extensive field data analyzed to identify the controls of preferential brook trout spawning sites. The data has not only been obtained for this particular study but represents a synthesis of newly acquired and existing data. However, the different data types as well as their location and timing of the collection make it hard to follow the story. My impression is that this is not because of the data as such but largely how it is presented. First I would have expected a rigorous evaluation of the factors that characterize the three preferred spawning sites such as EC, temperature

and oxygen and GW discharge and how these conditions are different from the other GW discharge sites which are not used for spawning. However, there is no synthesizing figure or table or section where I can learn about this. The results section is mostly a description of the individual results of the different methods. The second part of the paper should then look into the regional geological setting that ultimately determines the conditions at the spawning sites. I suggest the restructure the paper, clearly following the objectives the authors state in their introduction. They provide an excellent guideline for the entire paper. Additionally, many figures lack spatial reference or use non-unique references. For example in Fig. 4 and Fig. 7 the x-axis seem to start it different locations. Figure 5 has no x-axis at all. Please see also my specific comments below. Overall I found it very hard to keep track of which measurements have been conducted where and when.

Specific comments:

Figures: Generally, I recommend using consistent symbols and consistent spatial reference. Otherwise it is hard to recognize spatial setup. E.g.: in figure 6 spawn zones 1 2 3 are blue, red and green; figure 7: red, red blue; figure 8: blue green purple. Figure 1: Please indicate where b and c are located in a Figure 4: The decreasing order of the x-axis is confusing Figure 5: x-axis missing, What is Loc 15? Figure6: Would it harm to show boxplots instead of the time series? The exact timing of the variations does not seem to play a role. Figure 7: x-reference seems different from Figure 4 l.128: maybe explain which methods are lumped under the term 'geophysical remote sensing' l. 156-160: These objectives are absolutely reasonable but they are not reflected in the structure of the results or the discussion section. Just an example: the discussion starts with heat tracing. Why? It should be about the Spawing sites l. 508: The effects of shear stress and bed material seem to be important. So I wonder why this has been been considered as factors controlling spawning site preference. Both can be measured.

[Figure]

693, 2018.

---

## Author Comment (AC1) · 5 Mar 2018

We thank the Reviewer for their broad and helpful analysis of this manuscript. The public comment period for HESS-D extends until March 15, but it seemed prudent to develop a roadmap for our revision now, in case the Reviewer has any further specific feedback that could help us shape a more clear paper. After the comment period ends, we will also follow up with a detailed record of revisions made to an updated version of this manuscript. It is clear the Reviewer recognizes the extensive methodology and expertise that went into this study of why certain groundwater discharge zones host re-

peated trout spawning, while a multitude of other focused discharges within the same stream system do not. As you note, the inclusion of these various interdisciplinary methods in one paper is challenging, and we need to do a better job at clarifying the underlying story. This can start with an updated title, for which we suggest: "Hydrogeochemical Controls on Brook Trout Spawning Habitat in a Coastal Stream". This new title puts the emphasis on understanding the structural controls on why some discharges are oxygen rich along this coastal stream, yet most are oxygen poor. Trout utilize the former for spawning, as the eggs need dissolved oxygen to survive and properly develop. This relationship between trout spawning and bed sediment oxygen is already known, and discussed in several of the references we cite, so that point alone is not the focus of this research. In this study we could have sampled the 40+ focused groundwater discharges (identified with fiber-optic heat tracing) for parameters such as DO and EC, and compared to the n=3 that have been observed to repeatedly host trout redds. That could likely have resulted in the kind of pore water chemical dataset for which it would be appropriate to do various statistical analysis to indicate trout chemical preferences in a more systematic way than we have done. Instead, which was not made clear enough in the initial submitted text, we are operating under the assumption that for trout to use groundwater discharge zones for spawning shallow pore water must be oxic. Our main goals are then to use our unique multidisciplinary toolkit to understand why those specific groundwaters have dissolved oxygen, and how discharge patterns develop at the meter-scale. As the reviewer is likely aware, many statistically-based ecological studies result in empirical relationships between fish and habitat attributes, but not necessarily process-based understanding that can be readily transferable to other systems and future scenarios. We feel our study is valuable in showing trout in this coastal stream directly utilize discharges on meander bends that cut into mineral soils, as depicted in Figure 1a. This conceptual understanding, supported by various data types and not known previous to this study, is powerful as it can directly be used to guide stream restoration and to identify possible spawning zones in other coastal streams using surficial soils maps and high-resolution elevation data such as LIDAR.

Further, we show how the physical discharge of groundwater creates slumped alcoves outside of the main river flow that may offer additional favorable aspects for redd formation, such as reduced bed shear stress. We agree that revision of the main stated objectives is in order to best frame the motivation and results of this study. We suggest: 1. Identify preferred brook trout spawning locations, and determine if they are directly associated with the discharge of oxic groundwater through interface sediments. 2. Develop a hydrogeochemical understanding of trout-preferred groundwater discharge zones that can aid in their identification in other less-studied systems. The revised focus of Object 1 puts the emphasis on understanding if/how trout appear to use the discrete spatial zonation of focused groundwater discharges. Through considerable effort, we show through the modeling of multi-year summer bed temperature records that groundwater fluxes are highest directly at the cut bank margin, and fall off strongly over just a few meters toward the stream channel, as bed oxygen is also reduced. We use ground penetrating radar to map bed peat deposits that offer a processes-based explanation for both reduced upward water flux (low hydraulic conductivity) and dissolved oxygen (carbon source). We agree with the bulk of the Reviewers specific comments, and will make appropriate edits to address them. In regards to the term "short circuit discharge", although it may sound informal, this was introduced by the highly cited Conant 2004 paper to describe groundwater discharge in a similar hydrogeologic setting with low-K organic lenses. Again, we thank the Reviewer for the thoughtful and helpful review, and look forward to submitting a more clear and useful manuscript as a result.

---

## Author Comment (AC2) · 5 Mar 2018

Thank you for your review of this manuscript. As mentioned in our reply to Reviewer #1, the public comment period for HESS-D extends until March 15. However, it seemed prudent to develop a roadmap for our revision now, in case the Reviewer has any further specific feedback that could help us shape a more clear paper. After the comment period ends, we will also follow up with a detailed record of revisions made to an updated version of this manuscript. As you note, the inclusion of various interdisciplinary methods in one paper is challenging, and we need to do a better job at clarifying the

underlying goals and story. This can start with an updated title, for which we suggest: "Hydrogeochemical Controls on Brook Trout Spawning Habitat in a Coastal Stream". This new title puts the emphasis on understanding the structural controls on why some discharges are oxygen rich along this coastal stream, yet most are oxygen poor. Trout utilize the former for spawning, as the eggs need dissolved oxygen to survive and properly develop. This relationship between trout spawning and bed sediment oxygen is already known, and discussed in several of the references we cite, so that point alone is not the focus of this research. We realize this was not made clear in the previously submitted draft, particularly by our original Objective 1 which indicated a larger statistical analysis of discharge zone chemistry. Instead we focus on a subset of discharge zones that trout directly utilize to move beyond shallow streambed chemistry and determine their larger structural controls. The strengths of this paper are in generating transferable hydrogeological understanding of trout habitat. Figure 1a displays the conceptual understanding developed during this multi-year study where trout utilized more localized groundwater flowpaths that remain oxygen-rich for spawning in discrete patches on meander bends. Perhaps as it appears early in the manuscript this model seems obvious, but it took the combined geophysical, thermal, chemical, and fish observational data to reveal this. We used fiber-optic heat tracing to spatially map 40+ focused groundwater discharge zones found along 2 km of streambed, and then sample a subset of the largest for shallow pore water chemistry for comparison to the n=3 locations where trout are known to repeatedly spawn. Although this discharge chemical data is presented in Table 1, the Reviewer is correct in that the spawn vs non-spawn discharges were not summarized and compared. We now understand this summary comparison is needed, and this will be included as new Table 2. In terms of the Figures lacking spatial reference, we can indicate directly on Figure 2 where the heat traces of Figure 4 were collected, and use the same common "downstream distance" from the upper end of the stream as Figure 7. This will be helpful to the reader. We could also name the various groundwater discharge zones by this same downstream distance reference- do you think that would be helpful? Right now they have a

numerical naming scheme developed in previous published hydrological research from this stream. The radar data Figure 5 does not have a downstream distance specified, as the speed the radar floated down river varies somewhat, so the data stream is not linear in space along the images. However major discharge zones were marked directly on the record during acquisition and those are shown in the Figure (eg "Loc 15"). This can be better explained. In regards to Figure 6, which shows thousands of individually modeled vertical seepage fluxes, we tried box plots as the reviewer suggested. However, removing the time component does not allow direct comparison to the time variable discharge record that may specifically impact more local groundwater flowpaths (rather than regional). We will revisit these plots for clarity. Overall we will strive to clarify which methods were used where, when, and for what purpose. We thank the Reviewer for their time invested to help us make this a better and more useful paper.

---

## Referee Comment (RC3) · Anonymous Referee #3 · 13 Mar 2018

This paper presents a very interesting and important study on revealing what variables are important to fish for choosing their favorite spawning sites. There is a wide variety of data collected and presented in the manuscript, which provides a solid foundation for making this work a highly valuable contribution to the scientific community and to decision makers. However, the presentation of the paper could be improved by making it clear the linkages between what is observed and what can be concluded. Here are a few general comments for the authors to consider:

1. On the title, it sounded very exciting but it didn't come across to me why this is

called "working backwards" after I read the entire manuscript multiple times. So I would suggest to either change the title or emphasize in the paper why this is considered "working backwards" compared to existing practices.

2. It was very hard to follow all the data on a spatio-temporal map as they were taken at different places and at different times. It would be helpful to orient the readers where and when all those measurements were taken using a plot or table. It would be critical for this study to present the time window that is important for the fish habitat and whether the measurements were taken during that window. If they don't overlap, then explanations on why those measurements are relevant and useful would be needed.

3. It was not clear when the drivepoint and minipoint samples were taken. If they were taken during the time when the fiber optic and other temperature profiles were deployed it would be helpful to indicate the sampling time on the temperature and seepage plots.

4. The impacts of hydrogeologic properties on the seepage and DO concentration could be demonstrated using a simple flow model, which could strengthen the paper substantially in linking all the valuable data together and generate useful insights.

A few other specific comments:

1. The locations of the fiber optic measurements are not found in Figure 2C.

2. most of the readers would need more information on how to read the GPR images.

3. why there is no GPR information around spawn3?

4. On Figure 6, not all lines show up on every plot, so more explanation is needed. The two legend boxes were meant to be one? x-axis with actual dates might work better for readers than the ordinal day.

5 . Figure 8, it might be helpful to plot 1-to-1 scatters for colocated specific conductivity and DO.

[Figure]

693, 2018.

---

## Author Comment (AC3) · 15 Mar 2018

We warmly thank the Reviewer for the careful attention to our submitted manuscript, for the overall favorable impression of the work, and for the constructive suggestions. Through the combined 3 reviews we have identified several common themes that will improve the clarity and impact of this presentation. In regards to your 4 main points:

1. Our intention with the "working backwards from streambed thermal anomalies" title was to indicate how our work moves beyond measuring water fluxes and dissolved

chemistry at the streambed interface, and into the source aquifer to develop a physical transport-based understanding of why certain groundwater discharge zones had favorable characteristics for trout spawning. In this case "moving backwards" is from a groundwater discharge flowpath perspective (moving upgradient from the discharge interface), but we realize this meaning is somewhat opaque. We suggest an updated title: "Hydrogeochemical Controls on Brook Trout Spawning Habitat in a Coastal Stream". This title plays on the strengths of this study, which are to illuminate how local river geomorphology, sediments, and flow groundwater flowpaths interact to generate oxygenrich interface discharge zones in predictable positions along the reach.

2,3. A large panel will be added to Figure 2 to better show where and when the complimentary chemical, physical, geophysical, and temperature measurements were made for this study. The varied measurement timing will also be better clarified in the main text.

4. A simple reactive flow model could be generated to demonstrate uptake of oxygen around DOC sources (peat lenses) in the streambed, which differs from direct discharge from the mineral soil aquifer at meander bends. However, all three Reviewers commented on the sometimes overwhelming/confusing range of methods used here, and even the inclusion of a simple numerical model may not be a net positive to the manuscript readability. There are other empirical and model-based studies that have shown the relationship of DO uptake around in-situ DOC sources in sediment-water interface media, compared to DOC-poor media, and this referencing will be improved to support our interpretation.

---

## Author Response (AR1)

**Review #1 author responses**

We thank the Reviewer for their broad and helpful analysis of this manuscript. It is clear the Reviewer recognizes the extensive methodology and expertise that went into this study of why certain groundwater discharge zones host repeated trout spawning, while a multitude of other focused discharges within the same stream system do not. As you note, the inclusion of these various interdisciplinary methods in one paper is challenging, and we needed to do a better job at clarifying the underlying story. This starts with an updated title, for which we suggest: "*Hydrogeochemical Controls on Brook Trout Spawning Habitat in a Coastal Stream*". This new title puts the emphasis on understanding the structural controls on why some discharges are oxygen rich along this coastal stream, yet most are oxygen poor. Trout utilize the former for spawning, as the eggs need dissolved oxygen to survive and properly develop. This relationship between trout spawning and bed sediment oxygen is already known, and discussed in several of the references we cite, so that point alone is not the focus of this research. In this study we could have sampled the 40+ focused groundwater discharges (identified with fiber-optic heat tracing) for parameters such as DO and EC, and compared to the n=3 that have been observed to repeatedly host trout redds. That could likely have resulted in the kind of pore water chemical dataset for which it would be appropriate to do various statistical analysis to indicate trout chemical preferences in a more systematic way than we have done. Instead, which was not made clear enough in the initial submitted text, we are operating under the assumption that for trout to use groundwater discharge zones for spawning shallow pore water must be oxic.

We agree that revision of the main stated objectives is in order to best frame the motivation and results of this study. These are now stated as:

*1. Identify repeat brook trout spawning locations, and determine if they are directly associated with the preferential discharge of groundwater through interface sediments.*

*2. Develop a hydrogeochemical characterization of trout-preferred groundwater discharge zones that can aid in their identification in other less-studied systems and potential inclusion in stream habitat restoration efforts.*

As the Reviewer indicates, the different types of data collection used for this study were at times hard to follow in the original version. We have responded with a restructuring of the manuscript around themes, such as "*Spatial mapping of preferential groundwater discharges*" and "*Visualizing streambed sediment geologic structure*", and then divide the descriptions of the specific methods and results under these themes. We feel this will help readers better follow the main threads of the research presentation through the paper.

Main goals revolve around the use our unique multidisciplinary toolkit to understand why those specific groundwaters have dissolved oxygen, and how discharge patterns develop at the meter-scale. As the reviewer is likely aware, many statistically-based ecological studies result in empirical relationships between fish and habitat attributes, but not necessarily process-based understanding that can be readily transferable to other systems and future scenarios. We feel our study is valuable in showing trout in this coastal stream directly utilize discharges on meander bends that cut into mineral soils, as depicted in conceptual Figure 9. This conceptual understanding, supported by various data types and not known previous to this study, is powerful as it can directly be used to guide stream restoration and to identify possible spawning zones in other coastal streams using surficial soils maps and high-resolution elevation data such as LIDAR. Further, we show how the physical discharge of groundwater creates slumped alcoves outside of the main river flow that may offer additional favorable aspects for redd formation, such as reduced stream velocities.

The updated Table 1 is now reorganized by theme (eg streambed groundwater discharge zones vs near-bank spawn zones) and mean data are shown for each species. Newly added chemical and isotope data are shown in Table 2, and the vertical groundwater flux rates in Figure 6 have been converted to box plots so streambed seepage and spawn sites can be more easily compared.

We have now also put all data collection points on a common spatial reference scale: distance from an upstream bridge crossing/fish ladder. This is now reflected in all Figures, including Figure 4, which before had (in a confusing way) started with spatial reference at an arbitrary mid-stream point.

Specific comments
1. Line 47: Using the word "preferred" is problematic because this paper does not actually statistically analyze preference which would require an explicit comparison to what is available. I recommend removing the word "preferred" from the manuscript.
We agree. We now use the terminology "repeat spawning zones" to directly correspond with the annual observations of trout behavior (physical identification of spawning trout and geolocation of dropout PIT tags).
2. The manuscript refers to 10 years of observation that have gone into counting and mapping redds and spawning behavior, but these data (and sample sizes, etc.) are never presented. Perhaps the authors refer to a manuscript that published these data. In any case, it would be better to temper this kind of wording so that the readers won't be expecting some kind of explicit analysis of these data.
The wording in this section has been revised to explicitly state what type of trout spawning observations were made
3. Line 58: In many places, words and phrases are used that are colloquial and informal (e.g., short circuit, dropout, choked, etc.). Although I understand what the authors mean, many readers would not understand these phrases, so it would be best to go through the whole manuscript and eliminate all of this informal language.
We have attempted to revise the manuscript with an eye for informal language (eg "choked" was removed, among other phrases) but "short circuit discharge" and "dropout" PIT tags are known and accepted terms. For example short-circuit flowpaths are described in detail in the highly-cited Conant Jr. (2004) paper.
4. These are minor issues, but we refer to "coldwater and warmwater fishes" not "cold-water fish. It's just convention that this wording is recommended by the fisheries community (American Fisheries Society). In this sense, it is different from "cold-water seep", for example, which is a hydrologic
rather than ecological feature. Also, please be careful about referring to refugia, which
is the plural form of refuge or refugium. Thus, we don't write "refugias".
Instances of "cold-water fish" and "refugia" have been revised
5. Line 182: Shouldn't this be in hectares? The units have been converted to hectares
6. Line 193: The word "niche" does not have a scale per
se, so it isn't appropriate to use it this way. Be more explicit about what scale you are
writing about.  "niche" has been dropped here
7. Line 236: No information is provided on the number of fish that were
tagged. As outlined in my general comments, it would be better to keep the discussion
of fish general.  We have now tried to do that in the revised section regarding trout observations
8. Line 343: Can you cite the previous work on the distribution of brook
trout spawning? Otherwise, this can't really be presented as a result because we have
no data to evaluate in relation to this statement.  This is just a summary statement for the
findings of this work, not reference to previous work. We can see how this was confusing, and
have deleted the statement.
9. Figure 1b: This is really too small to
examine and appreciate. It's really impressive, but honestly the dots are too busy and
crowded.  We agree. The stream-scale FO-DTS data has now been broken out and enlarged in
the new Figure 1
10. Figure 7: The red and blue symbols are really hard to see on the graph
because they overlay one another. Can you jitter them a little so that it is easier to see
them? Yes, the point is that they directly overlay (similar chemistry), which of course make the
background symbol harder to see. The background point is enlarged so that all 3 symbols are
visible.
**Review #2 author responses**
The paper presents an extensive field data analyzed to identify the controls of preferential
brook trout spawning sites. The data has not only been obtained for this particular
study but represents a synthesis of newly acquired and existing data. However, the
different data types as well as their location and timing of the collection make it hard
to follow the story.
Thank you for the time invested in this review. We agree that the original submitted manuscript
was too hard to follow. Our group was so familiar with the site and study after several years of
work that we did not appreciate how confusing it would be for the reader to piece together all of
the various methods and results. With a fresh eye, and the helpful feedback from the first round
of review, we have revised the title, main objectives, and overall organization of the paper. The
"story" is now based around themes such as "*Spatial mapping of preferential groundwater*
*discharges*" and "*Visualizing streambed sediment geologic structure*", and then divide the
descriptions of the specific methods and results under these themes. We feel this will help
readers better follow the main threads of the research presentation through the paper.
My impression is that this is not because of the data as such but
largely how it is presented. First I would have expected a rigorous evaluation of the
factors that characterize the three preferred spawning sites such as EC, temperature and
oxygen and GW discharge and how these conditions are different from the other
GW discharge sites which are not used for spawning. However, there is no synthesizing
figure or table or section where I can learn about this.

The original Table 1 showed the SpC and DO data from various spawn sites and sampled
streambed groundwater discharge sites, but the data were organized by year of data collection
instead of theme, making the Table difficult to interpret. The updated Table 1 is now reorganized
by theme (eg streambed groundwater discharge zones vs near-bank spawn zones) and mean
data are shown for each species. We have added new Table 2 that includes water quality data
and stable water isotopes to further show distinguishing characteristics of repeat spawning
zones.
The results section is mostly
a description of the individual results of the different methods. The second part of the
paper should then look into the regional geological setting that ultimately determines
the conditions at the spawning sites. I suggest the restructure the paper, clearly following the
objectives the authors state in their introduction. They provide an excellent
guideline for the entire paper.
As mentioned above, the paper has been reorganized by theme instead of method, which
makes for a better flow of the main story.
Additionally, many figures lack spatial reference or use
non-unique references. For example in Fig. 4 and Fig. 7 the x-axis seem to start it different
locations. Figure 5 has no x-axis at all. Please see also my specific comments
below. Overall I found it very hard to keep track of which measurements have been
conducted where and when.
We have now put all data collection points on a common spatial reference scale: distance from
an upstream bridge crossing/fish ladder. This is now reflected in all Figures, including Figure 4,
which before had (in a confusing way) started with spatial reference at an arbitrary mid-stream
point.
Specific comments:
Figures: Generally, I recommend using consistent symbols and consistent spatial reference.
Otherwise it is hard to recognize spatial setup. E.g.: in figure 6 spawn zones 1
2 3 are blue, red and green; figure 7: red, red blue; figure 8: blue green purple.
Figure 6 has been changed to box plots as recommended by the Reviewer. Figure 8 shows
data at high spatial resolution in the shallow streambed, and the symbology was changed as to
be less-confusing when following Figure 7.
Figure
1: Please indicate where b and c are located in a Figure 4: The decreasing order of
the x-axis is confusing
The length of this FO-DTS deployment is now shown in Figure 2a, and the x-axis is now shown
in total stream length from the common upstream point
Figure 5: x-axis missing, What is Loc 15?
Radar collection time is now shown on the x-axis. This cannot readily be converted to a linear
spatial scale due to differential velocity of the river with distance, but the specific points of
groundwater discharge zones of interest were directly marked in the record, as is now shown on
the Figure.
Figure6: Would it
harm to show boxplots instead of the time series? The exact timing of the variations
does not seem to play a role.
We have made this change
Figure 7: x-reference seems different from Figure 4 These Figures now have a common spatial
reference
l.128: maybe explain which methods are lumped under the term 'geophysical remote
sensing' l. 156-160: These objectives are absolutely reasonable but they are not reflected in the structure of the results or the discussion section. Just an example: the
discussion starts with heat tracing. Why? The paper has been reorganized by theme as
discussed above
It should be about the Spawing sites l. 508:
The effects of shear stress and bed material seem to be important. So I wonder why
this has been been considered as factors controlling spawning site preference. Both
can be measured.  Shear stress was not measured or estimated here, so this statement was
removed

**Review #3 author responses**

We would like to thank the Reviewer for the careful attention to our submitted manuscript, for
the overall favorable impression of the work, and for the constructive suggestions. Through the
combined 3 reviews we have identified several common themes that will improve the clarity and
impact of this presentation. In regards to your 4 main points:
1. On the title, it sounded very exciting but it didn't come across to me why this is called
"working backwards" after I read the entire manuscript multiple times. So I would
suggest to either change the title or emphasize in the paper why this is considered
"working backwards" compared to existing practices.
Our intention with the "working backwards from streambed thermal anomalies" title was to
indicate how our work moves beyond measuring water fluxes and dissolved chemistry at the
streambed interface, and into the source aquifer to develop a physical transport-based
understanding of why certain groundwater discharge zones had favorable characteristics for
trout spawning. In this case "moving backwards" is from a groundwater discharge flowpath
perspective (moving upgradient from the discharge interface), but we realize this meaning is
somewhat opaque. We suggest an updated title: "Hydrogeochemical Controls on Brook Trout
Spawning Habitat in a Coastal Stream". This title plays on the strengths of this study, which are
to illuminate how local river geomorphology, sediments, and flow groundwater flowpaths interact
to generate oxygen-rich interface discharge zones in predictable positions along the reach.
2. It was very hard to follow all the data on a spatio-temporal map as they were taken
at different places and at different times. It would be helpful to orient the readers
where and when all those measurements were taken using a plot or table. It would be
critical for this study to present the time window that is important for the fish habitat and
whether the measurements were taken during that window. If they don't overlap, then
explanations on why those measurements are relevant and useful would be needed.
All data have now been put on a common spatial scale (distance from upstream bridge/fish
ladder), and all data collection sites are now shown in Figure 2. Measurements were all
collected in summer, which should represent an end member for (higher) DO consumption in
the streambed based on temperature-related kinetic rates. Extensive in-stream data collection is
not possible during the spawning season as this would disturb the trout eggs in the shallow bed
of this critical naturally reproducing (eg not stocked) trout system.
3. It was not clear when the drivepoint and minipoint samples were taken. If they were
taken during the time when the fiber optic and other temperature profiles were deployed
it would be helpful to indicate the sampling time on the temperature and seepage plots.
The drivepoint and minipoint data were collected in 2014 and 2016 during the times of
temperature data collection. All sampled locations are now noted on Figure 2a.

4. The impacts of hydrogeologic properties on the seepage and DO concentration
could be demonstrated using a simple flow model, which could strengthen the paper
substantially in linking all the valuable data together and generate useful insights.
A simple reactive flow model could be generated to demonstrate uptake of oxygen around DOC
sources (peat lenses) in the streambed, which differs from direct discharge from the mineral soil
aquifer at meander bends. However, all three Reviewers commented on the sometimes
overwhelming/confusing range of methods used here, and even the inclusion of a simple
numerical model may not be a net positive to the manuscript readability. There are other
empirical and model-based studies that have shown the relationship of DO uptake around in-
situ DOC sources in sediment-water interface media, compared to DOC-poor media, and this
referencing will be improved to support our interpretation.
A few other specific comments:
1. The locations of the fiber optic measurements are not found in Figure 2C.
The spatial range of the FO-DTS measurements shown in Figure 4 are now shown in Figure 2.
2. most of the readers would need more information on how to read the GPR images.
Yes, radargrams do need to be interpreted with some instruction/experience. We have
highlighted the main reflectors (interpret as sand/peat interfaces) to help with this interpretation
without adding extensive additional text
3. why there is no GPR information around spawn3?
The instrument was not deployed in this area unfortunately
4. On Figure 6, not all lines show up on every plot, so more explanation is needed. The
two legend boxes were meant to be one? x-axis with actual dates might work better for
readers than the ordinal day. This has been changed to a box plot representation, as suggested
by Reviewer #2
5 . Figure 8, it might be helpful to plot 1-to-1 scatters for colocated specific conductivity
and DO. We tried this, but the result was not clear.

[revised manuscript text omitted]

---

## Author Response (AR2)

**Review #1 Author Response**

**Please see our specific revisions below listed in black text:**

The authors have addressed my major concerns. This is an excellent, comprehensive study, and the authors should be commended on assembling all the components into a manuscript that will be appreciated by a wide, interdisciplinary audience. I have no further recommendations for major changes. However, I have identified many stylistic issues and minor problems that need to be rectified in the text and especially in the figures. These recommendations and suggestions are listed in the specific comments below.

Thank you for the positive assessment. We have addressed specific suggestions as detailed below.

Specific comments

Line 68: I recommend not capitalizing "Unmanned Aircraft Systems". Check for other examples throughout the manuscript. done

Line 80: Check spelling of "Mathews". Corrected

Line 81: No need to capitalize "Northeastern". fixed

Line 83: Use "cold-water" as a modifier of "habitat". done

Line 120: Citations should be in chronological order. We now believe the references are formatted per HESS guidelines, throughout

Line 152: "PIT" needs to be defined at first mention. done

Line 226: The acronym "PIT" without the definition can be used here because it will have been described earlier in the manuscript. done

Line 307: Check "pore water were also collected" for subject and verb agreement. This sentence
was modified

Line 321: Specify the wavelength for the thermal IR camera. We added "thermal" infrared to
indicate the broader IR spectrum and the model number is listed

Line 326: Define "SpC" at first mention. done

Line 331: No hyphen is needed in "locally-recharged" (https://www.merriam-
webster.com/words-at-play/6-common-hypercorrections-and-how-to-avoid-them/hyphenating-
ly-adverbs). Check entire document for other examples of this kind of incorrect hyphen use (e.g.,
"ecologically-based", etc.). changes made

Line 356: The second sentence in paragraph is a typographical error. I suggest that the first
sentence be moved to the beginning of the following section titled "Observations regarding
repeat spawning locations". This sentence was deleted. We need to preserve an intro sentence or
two for each main section per USGS style, as not to have "hanging titles"

Line 379: Check "data was". It should be "data were". changed

Results: The authors are consistent in reporting their results in the present tense, but this is
awkward to me because the conditions and observations are in the past. We have chosen to use
the present tense here to be consistent with other recent publications from the group. It seems
this falls in the category of "authors choice"

Line 477: "Global Positioning System" does not need to be capitalized. Agreed, changed

Line 598: "over 10 yr+" could be written as "> 10 yr". this was changed

Table captions: It is awkward to begin these captions with a number. Also, I found these captions
(and those of the figures) to be written in a nonstandard manner that makes it difficult to quickly
determine the contents of the table that follows. Other manuscripts by Briggs et al. don't have
this problem, so the lead author can certainly make the necessary changes. The captions were
adjusted as to not start with dates

Figure 1: Number in scale bar is too small. No need to write "Legend". Can you use arrows to
point to the locations of your sites instead of using purple circles, which obscure the temperature
dots? The way the captions is written, it is too difficult to quickly determine what the sites
labeled with "GW" mean. Honestly, the caption does an inadequate job of describing what this
figure depicts. For example, the first sentence of the caption is a statement of the methods. See
other papers written by Briggs et al. for better examples of how captions should be written.
These temperature data were from DTS, correct? This needs to be stated in the caption. These
data are plotted in Google Earth, where it is not obvious how to change the scale bar size (this
would be easy in Arcmap. The scalebar has been now enhanced in post processing. The purple
circles have been replaced with arrows and the caption adjusted to be more clear and completer.

Figure 2: Need "MA" on inset map. The numbers in the scale bars are too small. Just draw a
simple black bar in a white box to indicate 200 m. This will need to be done in each panel
because the scales are different. No need to write "Legend"; make the text larger in the contents.
It's still too hard to read. Can you define the purple dots and orange circles in the legend? The
transect A-A prime is not mentioned in the caption. Remove it if it is unimportant. It would be
helpful to have rectangles on Figure 2a to show the extents of the insets (Figures 2b and 2c). The
color scheme is problematic because there is no natural progression of colors. This may seem
like a minor point, but it does help the reader when interpreting the range from high to low.
Finally, the writing of the caption is awkward and not standard with "Panel c)" written in a
grammatically incorrect style. Please other papers by Briggs et al. for correct examples of how
captions are written.

We have made these changes where practical. Scale bars were replaced as requested. Font size is
practically limited by the size of the Figure, but it has been enhanced throughout the legend. The
groundwater discharge symbols and spawn locations have been added to the legend, and
"legend" deleted.The transect is important to Figure 9, as described now in legend. The inset
boxes were attempted, but make an already busy Figure too cluttered. Color schemes are in some
part a personal preference of the reader. We experimented with several gradational schemes for this, but feel the current scheme allows the floodplain to most clearly be distinguished from the
valley wall, which is directly relevant to the main theme of the paper. The key for color vs
elevation is clearly labeled.

Figure 3: See other comments about how captions should be written. Other papers by Briggs et
al. have excellent examples of nicely written captions. The caption has been edited

Figure 4: This caption is written better, but it will help to use () around the panel letters. Note
that a single parenthesis should be used where there is no text to the left (i.e., in a numerical list):
() added

1)

2)

3)

or

a)

b)

c)

Figure 5: Please write captions in a standard manner as is appropriate for a scientific journal. The
first line of the caption is not a methodological statement; it should be a concise statement
describing the contents of the figure. Also, it would be very helpful to provide a more explicit
description of the distance on the x-axis, even if it is approximate. See other comments about
how parentheses should be used if they are enclosing parenthetical information. Caption edited.
We do not feel comfortable adding an approximate x-axis due to variable float speed of the
radar, but the locations of interest (spawn zones, discharge zones) were directly marked in the
record and explicitly shown here.

Figure 6: See comments for other figures on how captions should be written. Caption edited

Figure 7: This caption is better than the others, but the panel labels need be enclosed in
parentheses in the caption. OK

Figure 8: The caption is poorly written. For example, to begin a caption with "Based on..." not
appropriate. There is no reason to provide citations in this caption. Perhaps the authors
mistakenly used a rough draft for all of these captions because Briggs et al. in their other papers
do a fine job of writing captions. Also, what is the brown arrow in the green area? Can you
remove this brown arrow? This figure also needs an arrow showing the direction of stream flow.
It's a difficult figure to understand because it's a 2-D graph with a 3-D figure inserted within it. A
flow direction arrow over the stream would help. We assume you refer to Figure 9 here, and the
suggestions have been adopted

**Review #2 Author Response**

   **Please see our specific revisions below listed in black text:**

   The revised paper has been significantly improved from the previous version. There are only a
few minor edits that are needed:
Line 236: This sentence is hard to understand, please consider rephrasing. Yes we were missing
a word in this sentence, and that has been fixed.
Line 340-341: please use the correct format for dates. Dates were changed to European format
throughout
Line 358: this is not a complete sentence and it appears to be a duplicate section title. Yes this
was a duplicate title, and has been deleted
Line 530: change "course" to "coarse" change made

[revised manuscript text omitted]